# Learning Symbolic Rules for Reasoning in Quasi-Natural Language

## Abstract

Symbolic reasoning, rule-based symbol manipulation, is a hallmark of human intelligence. However, rule-based systems have had limited success competing with learning-based systems outside formalized domains such as automated theorem proving. We hypothesize that this is due to the manual construction of rules in past attempts. In this work, we ask how we can build a rule-based system that can reason with natural language input but without the manual construction of rules. We propose MetaQNL, a "Quasi-Natural" language that can express both formal logic and natural language sentences, and MetaInduce, a learning algorithm that induces MetaQNL rules from training data consisting of questions and answers, with or without intermediate reasoning steps. Our approach achieves state-of-the-art accuracy on multiple reasoning benchmarks; it learns compact models with much less data and produces not only answers but also checkable proofs.

## 1 Introduction

Symbolic reasoning—rule-based symbol manipulation—is a core component of human intelligence (Mercier & Sperber, 2017). It has been a core part of computer science research, and has achieved significant success in domains such as software verification (Darvas et al., 2005) and theorem proving (Kovács & Voronkov, 2013; McCune, 1997). However, such success has been restricted to domains amenable to rigid, precise formalization. It remains a challenge how to translate such success into "informal" domains such as reasoning with common sense knowledge and natural language input. Prior attempts to build rule-based systems, which rely on manually constructed rules, have achieved limited success and tended to produce brittle systems.

Deep learning provides an attractive alternative that can easily sidestep the question of representation. A deep network can be trained to perform a reasoning task by directly predicting the answer without explicit symbol manipulation (Clark et al., 2020). However, deep networks can require a large amount of training data and can suffer from poor generalization. More importantly, unlike symbolic systems, a deep network is a black box that is hard to interpret, inspect, and verify. Such lack of interpretability can be undesirable in certain applications, especially those critical to safety and security.

In this work, we ask how to build a rule-based system that reasons symbolically but can work with natural language and handle domains difficult to formalize. Such a system would perform reasoning by explicit symbol manipulation based on known rules, therefore more interpretable and verifiable, but at the same time flexible enough to handle natural language input.

At a glance, this may appear a large departure from the conventional wisdom that learning-based systems, particularly deep networks, are far superior to rule-based systems, as history has demonstrated repeatedly. However, we hypothesize that this conventional wisdom is incorrect because it assumes a false dichotomy between using learning and using rules; rule-based systems underperformed not because they were rule-based, but because it is difficult to construct rules manually. Further, we hypothesize learning rules from data is key to building effective rule-based systems, but it may require a different kind of learning than gradient descent.

The goal of this work is thus to develop a method that automatically learns symbolic rules from data to enable rules-based reasoning with natural language. This poses two main questions. First, what is the system of rules—the basic structures that define what symbols and manipulations allowed—such

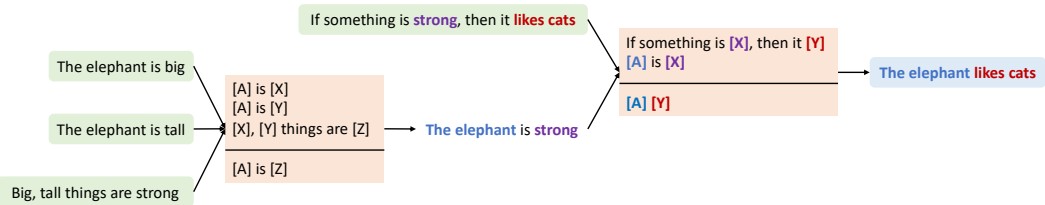

Figure 1: An example proof with 4 assumptions, 1 goal, and 2 rule applications. Each rule have multiple premises and one conclusion. Both the premises and the conclusion can have variables that bind to concrete sentences when the rule is applied.

that it is compatible with not only formal logic but also natural language? Second, what is the learning algorithm that induces a set of rules from training data?

In this work, we take initial steps toward answering both questions. We propose MetaQNL, a formal symbolic system we call a "Quasi-Natural Language", which is compatible with not only rigorous logical inference but also natural language expressions. We also propose MetaInduce, a learning algorithm that induces MetaQNL rules from training data that consists of questions and answers, with or without intermediate reasoning steps.

**MetaQNL: a Symbolic System in Quasi-Natural Language.** In MetaQNL, a *sentence* is a sequence of words and variables ("The elephant is [X]"). They also include ordinary English sentences without variables. A *rule* consists of multiple sentences as its premises ("The elephant is [X]", "If something is [X] then it [Y]") and one sentence as its conclusion ("The elephant [Y]"). When applying the rule in reasoning, variables are substituted with concrete sentences ([X] → strong, [Y] → likes cats). Therefore, rules capture abstract knowledge that is independent of specific instances—the above rule holds whether [Y] is "likes cats" or "is sleepy". Such abstraction is essential for reasoning in both humans and machines (Marcus & Davis, 2020).

Fig. 1 illustrates how sentences and rules are used in reasoning. Starting from known sentences (assumptions), we apply rules to derive new sentences until the goal is reached. At each step, we substitute the variables in a rule with concrete sentences. This process resembles Metamath (Megill & Wheeler, 2019), a formal language developed for formalizing mathematical proofs, where each step also consists of selecting a theorem and instantiating it with a suitable substitution. So we refer to the reasoning process as *theorem proving* and the result in Fig. 1 as a *proof*. It is worth noting that reasoning in MetaQNL is interpretable by design: it is transparent about what rules are assumed; it produces not only an answer, but also a proof that can be mechanically checked against the rules.

Assumptions and the goal are usually given when applying our system to a specific task. To solve the task, two issues remain: (1) *Rule induction*: What is the set of rules? (2) *Theorem proving*: How to apply the rules to find a proof? Theorem proving has been studied extensively in classical AI (Robinson & Voronkov, 2001) and more recently with deep learning (Alemi et al., 2016; Yang & Deng, 2019; Bansal et al., 2019), and we can adapt existing algorithms such as forward chaining and backward chaining (Russell & Norvig, 2002). In this work, we simply use existing provers and focus instead on the more challenging problem of rule induction.

**MetaInduce: an Algorithm to Learn MetaQNL Rules.** Rule induction can be formulated as a discrete optimization problem that seeks a minimum set of rules that are consistent with training examples. Note that it is important to seek a small number of rules because we always have a trivial solution that consists of one rule per example but is unlikely to generalize. This optimization is challenging due to the discrete, combinatorial search space.

We introduce MetaInduce, a general method for learning MetaQNL rules. It encodes the problem as a maximum satisfiability (MAX-SAT) problem, which can be solved efficiently by off-the-shelf solvers (De Moura & Bjørner, 2008). Our method consists of 3 steps. First, given a training example, a *rule proposer* proposes a set of concrete rules (rules without variables) as candidates. This set can be overcomplete and inaccurate. These rules are used to prove the example using existing provers such as forward/backward chaining. Second, we generate abstract rules from concrete rules via a symbolic procedure called anti-unification (Plotkin, 1970; Kutsia et al., 2014). Third, we encode the proof paths in MAX-SAT and solve for a subset of all rules using a MAX-SAT solver.

**Overview of Results.** We benchmark our method on 2 tasks: learning compositional instructions and logical reasoning. For learning compositional instructions, our method not only achieves 100% accuracy on two standard benchmarks: MiniSCAN (Lake et al., 2019) and SCAN (Lake & Baroni,

2018), but also recovers precisely the ground truth rules. For logical reasoning, our method achieves state of the art on the RuleTaker dataset (Clark et al., 2020). Compared to existing methods, our approach learns compact models with much less data, and produces not only answers but also checkable proofs. On RuleTaker, our approach learns a model that has only 2869 symbols but is competitive with a prior approach that uses a neural network with 11 billion parameters.

## 2 RELATED WORK

**Symbolic reasoning.** Symbolic reasoning has been studied extensively in classical AI, such as theorem proving (Kovács & Voronkov, 2013; Robinson & Voronkov, 2001). An open problem is to perform symbolic reasoning in domains without a natural formalization, such as natural images or texts. One common approach is to manually construct a formal system (e.g., based on first-order logic with manually defined functions and predicates), then perform semantic parsing to translate images or texts into formalized statements as input to a reasoning module operating in a clean formal world.

For example, to judge whether one statement implies another, Mineshima et al. (2015) use a semantic parser to convert both statements into higher-order logic (with predefined predicates), and then run an automated theorem prover. Semantic parsing is still far from reliable; therefore, researchers have developed techniques for learning it jointly with the reasoning module (Mao et al., 2018; Saparov & Mitchell, 2021; Dai et al., 2019; Li et al., 2020). In contrast, our approach does not require a semantic parser, because rules in MetaQNL are directly applicable to natural language.

Natural Logic (McAllester & Givan, 1993; MacCartney & Manning, 2007; Angeli et al., 2016) is a class of symbolic systems defined using the syntax of natural language, bypassing semantic parsing. Compared to our system, Natural Logic is more specialized because it is a specific logic committed to a set of predefined rules (namely seven set-theoretical relations), which restrict the type of reasoning it can perform to monotonicity reasoning (Icard III & Moss, 2014). In contrast, our system has no such restrictions because is not a specific logic but a meta-language with minimal structure such that it can instantiate various types of reasoning, just as MetaMath is a meta-language that can describe a variety of mathematical logics (Megill & Wheeler, 2019).

None of these works discussed so far learn rules from data; they instead use a predefined formal system that is already specialized and already encodes a substantial amount of prior knowledge. In contrast, MetaQNL is almost "knowledge-free" in the sense that it imposes the weakest possible structure on the permitted rules and lets the specific rules emerge from data through learning.

**Reasoning with neural networks.** Neural networks can perform "soft" reasoning in the space of continuous vectors without manipulating discrete symbols explicitly. Prior works have used transformer-based (Vaswani et al., 2017) language models for soft reasoning (Polu & Sutskever, 2020; Saha et al., 2020; Tafjord et al., 2020; Gontier et al., 2020; Talmor et al., 2020). Clark et al. (2020) finetune a pretrained transformer to classify whether the goal is provable from the assumptions, encoding them as sentences in a constrained natural language. Saha et al. (2020) and Tafjord et al. (2020) go one step further to generate proofs in addition to yes/no answers. Bostrom et al. (2021) generate conclusions from premises in unconstrained natural language.

Instead of using a generic transformer, researchers have also added inductive biases to the neural architecture. Many are inspired by symbolic reasoning and are often called *neuro-symbolic architectures*. Rocktäschel & Riedel (2017) introduce Neural Theorem Provers (NTPs). Given the assumptions and the goal in first-order logic, they use backward chaining to recursively construct a neural network. However, NTPs only work for formalized inputs and do not scale due to exponentially many proof paths in backward chaining. Weber et al. (2019) extend NTPs to natural language by extracting symbols from sentences using an off-the-shelf named entity recognizer. Minervini et al. (2020) make NTPs more scalable by dynamically pruning unpromising proof paths in backward chaining.

Researchers have also attempted to embed symbolic structures such as logic formulas into continuous vectors while preserving logical operations (Grefenstette, 2013; Kathryn & Mazaitis, 2018; Lee et al., 2016; Schlag et al., 2019). For example, tensor product representations (TPRs) (Smolensky, 1990) represent symbols as tensors and perform variable binding/unbinding via tensor operations such as inner/outer product. Dong et al. (2018) propose Neural Logic Machines (NLMs)—a neuro-symbolic architecture based on continuous approximation of logical inference. Predicates are represented as tensors; rules are neural operators that map tensors to tensors.

Cingillioglu & Russo (2020) propose an end-to-end neural architecture called unification networks to learn rules with variables from concrete examples. However, their system of rules is significantly less general than ours: their variables can only bind to a single word, whereas our variables bind to arbitrary sentences. In addition, their system does not support multistep chained reasoning. All reasoning is done in a single step: producing a conclusion in the form of an answer ("yes/no", a number, etc.) given a number of premises consisting of a question and a set of supporting facts.

Unlike these prior works, we learn symbolic rules instead of weights in a neural network. Further, during inference, we generate symbolic proofs whose correctness with respect to the induced rules is guaranteed and can be mechanically checked. Saha et al. (2020) and Tafjord et al. (2020) also generate proofs, but their proofs are natural language texts whose correctness is neither guaranteed nor mechanically checkable—their approach trains neural networks to directly predict both answers and proofs, but does not expose a system of rules against which a proof can be checked.

**Rule induction.** Inductive logic programming (ILP) learns rules in first-order logic programs such as Prolog and Datalog (Plotkin, 1972; Muggleton, 1991; Cropper & Dumančić, 2020). Extending it to natural language is non-trivial—partially due to the need for a predefined ontology of objects and predicates, as well as a perfect semantic parser, both of which are infeasible. Unlike ILP, we learn rules in MetaQNL, which can express not only logic programs but also natural language sentences. And our experiments show that MetaQNL can solve tasks that are not easily solvable by ILP.

For learning rules, our MetaInduce algorithm draws inspiration from existing ILP approaches. They encode proofs as either a boolean satisfiability problem solvable by off-the-shelf SAT solvers (Raghothaman et al., 2019) or a differentiable function amenable to gradient descent (Yang et al., 2017; Evans & Grefenstette, 2018; Si et al., 2019). Compared to these approaches, our rule space is different and more complex. Our rules consist of sentences with variables, whereas rules in ILP are typically Horn clauses in first-order logic. Further, ILP often imposes strong syntactic constraints on what rules are valid, e.g., using rule templates (Evans & Grefenstette, 2018; Raghothaman et al., 2019), or restricting to binary predicates (Evans & Grefenstette, 2018). These constraints are critical to good performance but are domain-dependent and difficult to get right (Cropper & Dumančić, 2020). Over-constraining the rule space makes the system less expressive, less generally applicable, and more brittle in the presence of noise. Another difference is that we minimize the number of rules in order to generalize, which is unnecessary for ILP due to stronger syntactic constraints.

Our space of rules includes a rich hierarchy from abstract rules to concrete rules, making the search space much larger. In contrast, most ILP works assume function-free first-order logic such as Datalog. Their variables can only be instantiated with concrete entities, making their rule space much simpler.

RNNLogic (Qu et al., 2021) learns first-order rules for knowledge base completion. They generate rules using RNNs, which is feasible because they require that rules can be expressed as a sequence of predicates. The strong syntactic constraint makes it less suitable for more general reasoning. Beyond first-order logic, Nye et al. (2020) learn rules for a string rewriting system. MetaQNL is more general because it can be applied to not only string rewriting but also other forms of reasoning (see Sec. 5).

## 3 MetaQNL: a Symbolic System in Quasi-Natural Language

MetaQNL is quasi-natural because it has a formal syntax compatible with natural language. Like in natural language, a *sentence* in MetaQNL is simply a sequence of tokens. There are 3 different types of tokens—words, variables, and special symbols. Taking the sentence "`$FALSE$ The elephant likes [X]`" as an example, "`The`", "`elephant`" and "`likes`" are words. MetaQNL treats words as symbols and does not assume any prior knowledge about their meaning. "`[X]`" is a variable—a placeholder that binds to concrete sentences in reasoning. "`$FALSE$`" is a special symbol. They are useful for encoding the structures of specific tasks, which will become more clear in Sec. 5. In this paper, we delimit special symbols with $. Sentences without variable are called *concrete sentences*, e.g., "`$FALSE$ The elephant likes cats`".

**Definition 1** (Sentence). *Let $\Sigma_w, \Sigma_v, \Sigma_s$ be vocabularies of words, variables, and special symbols respectively; they are disjoint and countable. Let $\Sigma = \Sigma_w \cup \Sigma_v \cup \Sigma_s$, then any $t \in \Sigma$ is a token. A sentence $s = (t_1, t_2, \ldots, t_n) \in \Sigma^+$ is a non-empty sequence of tokens. A concrete sentence is a sentence without any variable, i.e., $\forall i, t_i \notin \Sigma_v$.*

MetaQNL expresses permitted reasoning steps through *rules*. A rule has multiple sentences as its premises (`"The elephant [X]"`, `"If something [X] then it [Y]"`) and one sentence as the conclusion (`"The elephant [Y]"`). Intuitively, the conclusion should follow from the premises regardless of what values the variables take. *Concrete rules* are rules without variables.

**Definition 2** (Rule). *A rule takes the form of $p_1; p_2; \ldots; p_n \vdash c$, where $p_1, p_2, \ldots, p_n \in \Sigma^+$ are premises, and $c \in \Sigma^+$ is the conclusion. It is concrete if all premises and the conclusion are concrete.*

In reasoning, we instantiate rules with concrete rules by substituting all variables with concrete sentences. Given the rule $r_1 = $ `"The elephant [X]; If something [X], then it [Y] ⊢ The elephant [Y]"`, we can instantiate it with the substitution $\{$`[X]` $\rightarrow$ `is strong,` `[Y]` $\rightarrow$ `likes cats`$\}$, deriving the concrete rule $r_2 = $ `"The elephant is strong; If something is strong, then it likes cats ⊢ The elephant likes cats"`. In such cases, we say $r_1$ is more general than $r_2$, or vice versa, $r_2$ is an instance of $r_1$.

**Definition 3** (Substitution). *Let $\Sigma^+_{-s} = (\Sigma_w \cup \Sigma_v)^+$ be the set of sentences with only words and variables (without special symbols). A substitution $\sigma$ is a function from $\Sigma_v$ to $\Sigma^+_{-s}$. Substitutions can be extended to be functions on tokens, sentences, and rules. Given a token $t \in \Sigma$, applying the substitution $\sigma$ produces a sentence $\sigma t$.*

$$\sigma t = \begin{cases} \sigma(t) & \text{if } t \in \Sigma_v, \\ t & \text{if } t \notin \Sigma_v. \end{cases}$$

*Given a sentence $s = (t_1, t_2, \ldots, t_n)$, applying $\sigma$ produces $\sigma s = (\sigma t_1, \sigma t_2, \ldots, \sigma t_n)$. Given a rule $r = p_1; p_2; \ldots; p_n \vdash c$, applying $\sigma$ produces $\sigma r = \sigma p_1; \sigma p_2; \ldots; \sigma p_n \vdash \sigma c$.*

We are abusing notations to treat a token and a single-token sentence interchangeably. Also, $(s_1, s_2, \ldots, s_n)$ denotes concatenation when $s_i$ are sentences. Substitution is defined as a function on all variables $\Sigma_v$, but in practice it only involves a few. For example, the substitution $\sigma = \{$`[X]` $\rightarrow$ `is strong,` `[Y]` $\rightarrow$ `likes cats`$\}$ only involves two variables. In such cases, we think of it as being the identity function for other variables, e.g., $\sigma$`[Z]` $=$ `[Z]`. This convention makes it easier to composite substitutions as function composition.

As in the example before, applying a substitution to sentences/rules makes them more specific. It introduces a partial order among sentences/rules. It is straightforward to verify that the relation $\leq$ defined below is a partial order. We leave the proof to Appendix A.

**Definition 4** (Partial order among sentences and rules). *Let $s_1$ and $s_2$ be two sentences, $s_1$ is an instance of $s_2$ (denoted by $s_1 \leq s_2$) if and only if there exists a substitution $\sigma$ such that $s_1 = \sigma s_2$. In this case, we also say $s_2$ is more general than $s_1$. Similarly, given two rules $r_1$ and $r_2$, $r_1$ is an instance of $r_2$ (or $r_2$ is more general than $r_1$, denoted by $r_1 \leq r_2$) if and only if $\exists \sigma, r_1 = \sigma r_2$.*

A subtlety in the definition is judging whether two rules are equal. For a MetaQNL rule, premises are unordered, and variable renaming does not matter. In more jargonized words, rule equality is defined modulo premise reordering and $\alpha$-conversion.

In reasoning (Fig. 1), the prover is given a set of rules $\mathcal{M}$, multiple concrete sentences $A$ as assumptions, and one sentence $g$ as the goal. It iteratively instantiates concrete rules from $\mathcal{M}$ and applies them to generate a proof of $g$. Similar to Prolog, $g$ may have variables (`The elephant [X]`), and the prover succeeds if it proves any instance of $g$ (E.g., `The elephant likes vegetables`).

**Definition 5** (Proof). *A proof $P = (V, E)$ is a directed acyclic graph whose vertices $V$ are concrete sentences or concrete rules. For each concrete rule $r = p_1; p_2; \ldots p_n \vdash c \in V$, it must satisfy two conditions: (1) $r$ connects to its conclusion $c \in V$ via an edge $(r, c) \in E$; (2) For each premise $p_i$, we have $p_i \in V$ and $(p_i, r) \in E$. Besides these edges, there cannot be any other edge in $E$. Also, there can be multiple sentences without inbound edges (the proof's assumptions), but there is only one sentence without outbound edges (the proof's goal).*

**Definition 6** (Theorem proving). *Given a set of rules $\mathcal{M} = \{r_1, r_2, \ldots, r_k\}$, concrete sentences $A = \{a_1, a_2, \ldots, a_n\}$ as assumptions, and a sentence $g$ as the goal. The theorem prover tries to find a proof $P$ such that: (1) $P$'s assumptions are $A$. (2) $P$'s goal is an instance of $g$. (3) Every rule $r$ in $P$ is an instance of a rule in $\mathcal{M}$.*

## 4 METAINDUCE: LEARNING METAQNL RULES FROM DATA

**Problem setup and loss function.** Rule induction is a machine learning problem where the model consists of rules rather than continuous weights. The problem setup is familiar: Given a training set $\mathcal{D}_{\text{train}}$ and a test set $\mathcal{D}_{\text{test}}$, the goal is to use $\mathcal{D}_{\text{train}}$ to find a model that performs well not only on $\mathcal{D}_{\text{train}}$ itself but also on $\mathcal{D}_{\text{test}}$. For MetaQNL specifically, the training set $\mathcal{D}_{\text{train}} = \{\mathcal{D}_{\text{train}}^+, \mathcal{D}_{\text{train}}^-\}$ consists of a set of *provable* examples $\mathcal{D}_{\text{train}}^+$ and a set of *unprovable* examples $\mathcal{D}_{\text{train}}^-$. They both contain training examples in the form of $(A_i, g_i)$, where $A_i$ is a set of assumptions and $g_i$ is the goal. A model $\mathcal{M}$ is consistent with a provable example $(A_i, g_i) \in \mathcal{D}_{\text{train}}^+$ if $g_i$ is provable from $A_i$ using rules in $\mathcal{M}$. Similarly, $\mathcal{M}$ is consistent with an unprovable example $(A_i, g_i) \in \mathcal{D}_{\text{train}}^-$ if $g_i$ cannot be proved from $A_i$. In other words, provable examples are positive examples demonstrating sound logical inference, whereas unprovable examples are negative examples demonstrating unsound inference.

Given only $\mathcal{D}_{\text{train}}$, we need to find a model consistent with as many examples in $\mathcal{D}_{\text{test}}$ as possible. However, it is not sufficient to optimize the consistency with training data, because there is a trivial model that performs perfectly in training but fails in testing—one rule per example. That is, given a example $(A_i, g_i) \in \mathcal{D}_{\text{train}}^+$, if $A_i = \{a_1, a_2, \ldots, a_k\}$, it is provable using the rule $a_1; a_2; \ldots; a_k \vdash g_i$.

Thus we need to penalize the model complexity. While other choices are possible, here we measure model complexity as the number of rules. We minimize a loss function that evaluates both model complexity and consistency with training data:

$$\mathcal{L}(\mathcal{M}) = |\mathcal{M}| - \lambda^+ \mathcal{N}(\mathcal{M}, \mathcal{D}_{\text{train}}^+) - \lambda^- \mathcal{N}(\mathcal{M}, \mathcal{D}_{\text{train}}^-), \tag{1}$$

where $|\mathcal{M}|$ is the number of rules; $\mathcal{N}(\mathcal{M}, \mathcal{D}_{\text{train}}^+)$ and $\mathcal{N}(\mathcal{M}, \mathcal{D}_{\text{train}}^-)$ are the number of provable/unprovable examples consistent with $\mathcal{M}$ respectively. $\lambda^+$ and $\lambda^-$ are hyperparameters controlling the trade-off between three terms.

The optimization problem is challenging. Given $\mathcal{M}$, even a single evaluation of $\mathcal{L}(\mathcal{M})$ is computationally expensive: $|\mathcal{M}|$ is trivial, but $\mathcal{N}(\mathcal{M}, \mathcal{D}_{\text{train}}^+)$ and $\mathcal{N}(\mathcal{M}, \mathcal{D}_{\text{train}}^-)$ require running the prover on all training examples. Furthermore, it is much harder to find the optimal $\mathcal{M}$ due to the combinatorial and non-differentiable search space. We introduce MetaInduce, a general method for learning rules in $\mathcal{M}$ by encoding Equation 1 as a maximum satisfiability (MAX-SAT) problem, which can be solved efficiently by off-the-shelf solvers.

---

**Algorithm 1:** MetaInduce: A general method for learning MetaQNL rules from data.

**Input** :Training data $\mathcal{D}_{\text{train}} = \{(A_i, g_i)\}_{i=1}^n$; $A_i$ is a set of assumptions; $g_i$ is the goal.
**Output** :A model $\mathcal{M}$ consisting of a set of rules

1   $\mathcal{M} \leftarrow \varnothing$
2   **for** $j \leftarrow 1$ **to** *num_epochs* **do**
3      **for** $i \leftarrow 1$ **to** $n$ **do**
4         candidates $\leftarrow$ `propose_rules`$(\mathcal{D}_{\text{train}}, i)$
5         `prove`$(A_i, g_i, \textit{candidates} \cup \mathcal{M})$
6      rules $\leftarrow$ `abstract_rules()`
7      $\mathcal{M} \leftarrow$ `prune_rules`(rules)

---

**Overview of MetaInduce.** MetaInduce is outlined in Algorithm 1. Similar to SGD for training neural networks, MetaInduce goes through the training data for several epochs; during an epoch, it processes one example per iteration. Given an example $(A_i, g_i)$ (either provable or unprovable), it first relies on a *rule proposer* for generating candidate rules that are concrete and potentially useful for proving $g_i$ from $A_i$. Then it runs an existing prover to search for proofs, using both the candidate rules and existing rules in the model. At the end of each epoch, MetaInduce abstracts all concrete rules used in the proofs into rules with variables. Then it performs rule pruning—selecting $\mathcal{M}$ as a subset of the rules minimizing the loss (Equation 1). Next, we explain each step in more detail.

**Rule proposal.** The rule proposer is dataset-dependent and allows incorporating prior knowledge about a particular task. However, a good rule proposer alone—if not embedded in MetaInduce—is not sufficient for learning rules. First, the rule proposer only generates concrete rules. It is up to MetaInduce to abstract them into rules with variables. Second, the rule proposer generates rules useful for a single training example, whereas MetaInduce learns rules useful for the entire dataset.

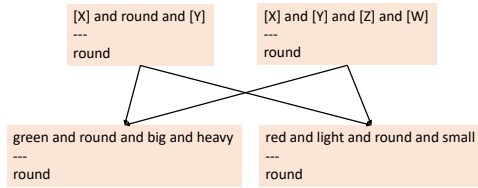
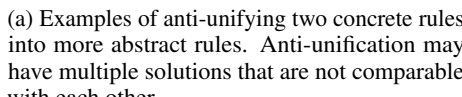
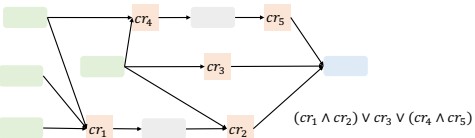

(a) Examples of anti-unifying two concrete rules into more abstract rules. Anti-unification may have multiple solutions that are not comparable with each other.

(b) Encoding as a boolean constraint a proof with 3 paths from assumptions to the conclusion. Each concrete rule $cr_i$ corresponds to a boolean variable. The proof is a disjunction of all paths; each path is a conjunction of concrete rules.

Third, the rule proposer does not have to be accurate. MetaInduce can reliably learn correct rules even if most candidate rules are wrong (see Sec. 5).

**Theorem proving.** Theorem proving in MetaQNL is relatively straightforward, thanks to existing algorithms such as forward/backward chaining. Forward chaining starts with the assumptions and applies rules to derive new sentences until the goal is reached. Conversely, backward chaining starts with the goal and applies rules in the reverse direction until all assumptions are satisfied. We implement forward chaining using the Rete algorithm for fast rule matching (Doorenbos, 1995) and the basic backward chaining algorithm from a standard textbook (Russell & Norvig, 2002). The prover returns proofs containing all different paths to the goal up to a predefined depth limit.

**Rule abstraction.** The proofs contain only concrete rules (Definition 5), and we have to generalize them into rules with variables. We use a symbolic procedure called *anti-unification* (Plotkin, 1970) to find general rules given concrete one. Given two rules $r_1$ and $r_2$, anti-unification attempts to find the most specific rule $r$ such that $r_1 \leq r$ and $r_2 \leq r$ (analogous to the lowest common ancestor of two nodes in a tree; see Fig. 2a for examples). It does so by recursively matching the beginning of two sentences. Please see Appendix B for details.

Let $\Gamma$ be the set of all concrete rules in the proofs. To augment $\Gamma$ with general rules, we iteratively anti-unify rules in $\Gamma$ and add the result back, until no new rule can be generated. We denote the result by $\Gamma'$, which has not only concrete rules but also their generalizations.

**Rule pruning with MAX-SAT.** Rule pruning selects $\mathcal{M}$ as a subset of $\Gamma'$ by encoding all proofs as a MAX-SAT problem, whose solution corresponds to a set of rules that approximately minimize the loss function in Equation 1. We encode each rule $r \in \Gamma'$ using a boolean variable (also denoted $r$). $r = 1$ means the rule should be included in $\mathcal{M}$. For any concrete rule $cr \in \Gamma$, we have an additional boolean variable $cr$. $cr = 1$ means $cr$ is necessary for proving the training examples. We impose 3 different types of constraints on these boolean variables:

- *Data consistency*: For the $i$th training example, its proof $P_i$ may have many paths from the assumptions to the goal, but the example is provable as long as at least one of them is valid. For provable examples (those in $\mathcal{D}_{\text{train}}^+$), we encode $P_i$ as a disjunction of proof paths. Each path is valid if and only if all concrete rules along the path are valid. So we encode a proof path as a conjunction of all $cr$ boolean variables it contains (see Fig. 2b). Analogously, for unprovable examples (those in $\mathcal{D}_{\text{train}}^-$), we simply take the negation of the previous boolean formula to encourage the absence of a valid proof. Finally, a good model is not necessarily consistent with every training example. So $P_i$ is encoded as a soft constraint with weight $\lambda^+$ or $\lambda^-$.

- *Model complexity*: To minimize the number of rules, we add a soft constraint $\neg r$ of weight 1 for each $r$ boolean variables. It encourages $r = 0$.

- *Rules instantiation*: Each concrete rule $cr$ must be an instance of a rule $r$. Let $r_1, r_2, \ldots, r_k \in \Gamma'$ be the set of all rules in $\Gamma'$ such that $cr \leq r_i$. $cr$ can be instantiated only if at least one of them is in the model. Therefore, we add a hard constraint $cr \rightarrow r_1 \vee r_2 \vee \cdots \vee r_k$.

Given a set of boolean constraints, each with a weight, a MAX-SAT solver finds an assignment of boolean variables to minimize the combined weights of violated constraints, which equals to Equation 1 for the specific constraints above. Therefore, running an off-the-shelf MAX-SAT solver on these constraints gives us a set of rules that minimizes our loss function.

## 5 EXPERIMENTS

We instantiate MetaQNL and MetaInduce on two tasks: learning compositional instructions in MiniS-CAN (Lake et al., 2019)/SCAN (Lake & Baroni, 2018) and logical reasoning in RuleTaker (Clark

et al., 2020). Not only does MetaInduce learn rules achieving state-of-the-art prediction accuracy on all 3 datasets, but it uses only a minor fraction of training data. Further, the rules recovered by MetaInduce match precisely with the ground truth rules of MiniSCAN and SCAN.

**MiniSCAN.** The data was introduced for studying few-shot learning of compositional instructions. For example, given "`dax → RED`", "`wif → GREEN`", and "`dax fep → RED RED RED`", we should learn "`wif fep → GREEN GREEN GREEN`". MiniSCAN consists of only 14 training examples (see Appendix C). Humans achieve an average accuracy of 84.3%, whereas state-of-the-art machine learning methods have reached 100% (Chen et al., 2020; Liu et al., 2020).

Each example in MiniSCAN consists of an input sequence $x$ and an output sequence $y$. For example, $x = $ `dax fep` and $y = $ `RED RED RED`. In training, we treat each input/output pair as a provable example $(A_i, g_i) \in \mathcal{D}_{\text{train}}^+$, with empty assumptions $A_i = \varnothing$ and the goal $g_i = $ "$x$ `$MAPS_TO$` $y$", e.g., "`dax fep $MAPS_TO$ RED RED RED`". In testing, we use "$x$ `$MAPS_TO$ [Y]`" as the goal, where `[Y]` is a placeholder to be filled by the prover. The prover succeeds if it proves a goal with any `[Y]`. We do not include any unprovable examples, i.e., $\mathcal{D}_{\text{train}}^- = \varnothing$.

We use a rule proposer independent of specific training examples. It generates all concrete rules with $\leq 2$ premises by combining the 14 sentences in the training data in all possible ways, leading to 1288 candidate rules. With backward chaining as the prover and Z3 (De Moura & Bjørner, 2008) as the MAX-SAT solver, MetaInduce successfully recovers all 7 ground truth rules of MiniSCAN and achieves 100% prediction accuracy. Here is one example of learned rules: "`[A] $MAPS_TO$ [B] ⊢ [A] fep $MAPS_TO$ [B] [B] [B]`". See Appendix C for more examples.

**SCAN.** As a standard benchmark for compositional generalization, SCAN is similar to MiniSCAN in format but much larger in scale. It consists of 21K examples of translating simplified natural language into action sequences. For example, "`jump → JUMP`", "`jump twice → JUMP JUMP`". State of the art has reached 100% accuracy on 4 different data splits: `simple`, `length`, `addprim_jump`, and `addprim_turn_left` (Liu et al., 2020; Nye et al., 2020; Chen et al., 2020).

We apply our method to SCAN similarly to MiniSCAN except for the rule proposer. SCAN is much larger; it is not feasible to generate all concrete rules exhaustively up to a certain number of premises. Therefore, we filter the rules using prior knowledge about compositional generalization: The meaning of a long sequence depends on the meaning of its subsequences. For example. "`jump $MAPS_TO$ JUMP ⊢ jump twice $MAPS_TO$ JUMP JUMP`" is a valid rule, since `jump` is a subsequence of `jump twice` (more examples in Appendix D). In contrast, "`look $MAPS_TO$ LOOK ⊢ jump twice $MAPS_TO$ JUMP JUMP`" is not a valid rule. Note that similar assumptions are also made in prior works (Nye et al., 2020; Liu et al., 2020).

Trained only on the 400 shortest examples, MetaInduce achieves 100% testing accuracy and recovers the 20 ground truth rules of SCAN on all 4 splits (`simple`, `length`, `addprim_jump`, and `addprim_turn_left`). Here is one learned rule: "`[A] $MAPS_TO$ [B]; [C] $MAPS_TO$ [D] ⊢ [A] after [C] $MAPS_TO$ [D] [B]`". Others are in Appendix D. The hyperparameter $\lambda^+$ is tuned on 1000 validation examples. The validation accuracy is fairly robust w.r.t. different $\lambda^+$ (Table 1).

Table 1: Validation accuracies on the `length` split of SCAN with different $\lambda^+$. $\infty$ means encoding data consistency as hard constraints.

| $\lambda^+$ | 0.32 | 0.64 | 1.28 | 2.56 | 5.12 | 10.24 | $\infty$ |
|---|---|---|---|---|---|---|---|
| #Rules learned | 16 | 17 | 20 | 20 | 20 | 20 | 20 |
| Accuracy | 85.9 | 90.3 | 100.0 | 100.0 | 100.0 | 100.0 | 100.0 |

**Logical reasoning on RuleTaker.** The RuleTaker dataset tests logical reasoning in synthetic English sentences. It consists of data examples similar to the one in Fig. 1. The original RuleTaker is generated with the closed-world assumption (CWA)—it assumes a sentence is false if it is not provable. Tafjord et al. (2020) introduces a version of RuleTaker with the open-world assumption (OWA). Under OWA, a sentence can be proved, disproved, or neither. We benchmark on the OWA version.

Some examples in RuleTaker are meant to be disproved: If "`The elephant is tall`" is true, then "`The elephant is not tall`" should be false. We add special symbols `$TRUE$` or `$FALSE$` before sentences, so that the previous example can be disproved using the rule "`$TRUE$`

Table 2: Answer predicting accuracies on the OWA version of RuleTaker. The model is trained on D5 or D3, and tested on D5 (proof depth $\leq 5$). Columns correspond to different proof depths within the test data. N/A means there is no proof since the test example can be neither proved nor disproved.

| Train | Model | N/A | 0 | 1 | 2 | 3 | 4 | 5 | All |
|-------|-------|-----|-----|-----|-----|-----|-----|-----|-----|
| D3 | ProofWriter | **99.9** | 100.0 | 99.3 | 99.7 | **99.2** | **99.1** | **98.8** | **99.6** |
|    | Ours | 99.4 | 100.0 | **100.0** | 99.7 | 98.9 | 98.9 | 98.6 | 99.4 |
| D5 | ProofWriter | **99.9** | 100.0 | 99.3 | 99.7 | 99.2 | 99.1 | 98.8 | 99.6 |
|    | Ours | 99.6 | 100.0 | **100.0** | **100.0** | **100.0** | **99.4** | **99.1** | **99.7** |

The elephant is tall ⊢ $FALSE$ The elephant is not tall". For each example to be proved, we add it to the set of provable examples $\mathcal{D}_{\text{train}}^{+}$ and its negation to unprovable examples $\mathcal{D}_{\text{train}}^{-}$. Conversely, for each example to be disproved, we add it to $\mathcal{D}_{\text{train}}^{-}$ and its negation to $\mathcal{D}_{\text{train}}^{+}$. For examples that can be neither proved nor disproved, we add itself and its negation to $\mathcal{D}_{\text{train}}^{-}$.

RuleTaker includes ground truth proofs providing concrete rules such as "$TRUE$ The elephant is tall ⊢ $FALSE$ The elephant is not tall" but not any abstraction that allows generalizing beyond the specific examples. Our rule proposer simply generates these ground truth concrete rules, whereas MetaInduce tries to learn general rules such as "$TRUE$ [X] is [Y] ⊢ $FALSE$ [X] is not [Y]". And we use simple heuristics for filtering invalid rules generated by anti-unification. Please see Appendix E for details and example rules generated by the rule proposer. All experiments are on machines with 0 GPUs, 32GB RAM, and four Intel Xeon Silver 4114 CPUs. We run MetaInduce for 5 epochs on a random subset of 10000 training examples, which takes about 20 hours. We use forward chaining as the prover and a depth limit of 7. The hyperparameters $\lambda^{+}$ and $\lambda^{-}$ are tuned on validation data.

We compare our method with ProofWriter (Tafjord et al., 2020)—a state-of-the-art method that also uses ground truth proofs. Following their setup, we test on D5 (a subset of RuleTaker with proof depths $\leq 5$) and train separate models on D5 and D3 (proof depths $\leq 3$). Training on D3 is for evaluating the model's generalization to longer proofs. Results are in Table 2. MetaInduce achieves state-of-the-art accuracy and is competitive with ProofWriter.

MetaInduce learns significantly more compact models with much less training data. For example, the model trained on D3 with $\lambda^{+} = \lambda^{-} = 1.28$ using only $14\%$ of the training data has only 79 rules and a total of 2869 symbols, but achieves a test accuracy of 99.4. In comparison, ProofWriter has an accuracy of 99.6 and is based on T5-11B (Raffel et al., 2020), which has 11 billion parameters.

The rules learned by MetaInduce enable reasoning that is not in the ground truth proofs. For example, it learns the rule "$TRUE$ if something [B] then it [C]; $TRUE$ [A] [B] ⊢ $TRUE$ [A] [C]". During inference, the rule is instantiated to "$TRUE$ if something be young then it be cold; $TRUE$ all furry thing be young ⊢ $TRUE$ all furry thing be cold". This pattern of reasoning makes sense but has never appeared in the grond truth proofs in RuleTaker.

# 6 LIMITATIONS AND OPEN QUESTIONS

First of all, our approach is far from mature. Substantial further development is needed for handling real-world natural language. Our experiments are neither large-scale nor real-world. They are small and synthetic, but serve as proof of concept for a very novel approach at an early stage.

MetaInduce does not yet scale to millions of training examples, which may be necessary to learn enough rules to handle the complexity of real-world natural language. The current bottleneck is anti-unification, which can be possibly addressed through better methods for generalizing concrete rules to rules with variables.

MetaInduce is a meta algorithm that permits many variations of its components. This provides many open questions and opportunities for integration with deep learning. For example, the rule proposer or theorem prover can be a deep network instead of a manually crafted heuristic.

In its current form, MetaQNL does not allow uncertainty, which is necessary for robustness to noisy input and for commonsense reasoning, where many statements are not categorical. How to introduce uncertainty to MetaQNL is an open research question. One possibility is to associate probabilities with sentences and rules, and further define how probabilities propagate through inference.

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

# A PARTIAL ORDER AMONG SENTENCES AND RULES

Here we prove that the $\leq$ in Definition 4 of the main paper is indeed a partial order relation.

**Definition 7** (Sentence length). *The length of a sentence $s = (t_1, t_2, \ldots, t_n) \in \Sigma^+$ is $length(s) = n$.*

**Lemma 1** (Substitutions are noncontractive). *Applying substitutions does not make a sentence shorter. In other words, for any sentence $s = (t_1, t_2, \ldots, t_n) \in \Sigma^+$ and substitution $\sigma : \Sigma_v \to \Sigma^+_{-s}$, we have $length(\sigma s) \geq n$. Further, $length(\sigma s) = n$ if and only if $\sigma$ maps all tokens in $s$ to sentences of length 1, i.e., $\forall i, length(\sigma t_i) = 1$.*

*Proof.* For any substitution $\sigma : \Sigma_v \to \Sigma^+_{-s}$ and variable $v \in \Sigma_v$, $\sigma(v) \in \Sigma^+_{-s}$ is a sentence. Therefore, for any token $t \in \Sigma$, $length(\sigma t) \geq 1$ (Definition 3). For any sentence $s = (t_1, t_2, \ldots, t_n)$, we have $length(\sigma s) = \sum_{i=1}^n length(\sigma t_i) \geq n$. And the equality holds if and only if $\forall i, length(\sigma t_i) = 1$. $\square$

**Theorem 1** (Partial order among sentences). *If sentence equality is defined modulo $\alpha$-conversion, then the $\leq$ in Definition 4 is a partial order among sentences. In other words,*

1. *$\forall s \in \Sigma^+, s \leq s$.*

2. *$\forall s_1, s_2 \in \Sigma^+$, if $s_1 \leq s_2$ and $s_2 \leq s_1$, then $s_1 = s_2$ modulo $\alpha$-conversion.*

3. *$\forall s_1, s_2, s_3 \in \Sigma^+$, if $s_1 \leq s_2$ and $s_2 \leq s_3$, then $s_1 \leq s_3$.*

*Proof.* We prove the 3 statements separately.

1. Let $\epsilon$ be the identity substitution mapping any variable to itself, i.e., $\forall v \in \Sigma_v, \epsilon(v) = v$. According to Definition 3, $\epsilon$ also maps any token to itself ($\forall t \in \Sigma, \epsilon t = t$), and therefore any sentence to itself ($\forall s \in \Sigma^+, \epsilon s = s$). Applying Definition 4, we have $\forall s \in \Sigma^+, s \leq s$.

2. Given two sentences $s_1 = (t_1, t_2, \ldots, t_n)$, $s_2 = (t'_1, t'_2, \ldots, t'_m)$ such that $s_1 \leq s_2$ and $s_2 \leq s_1$, there exist substitutions $\sigma, \varphi$ such that $s_1 = \sigma s_2$ and $s_2 = \varphi s_1$ (Definition 4). Applying Lemma 1 to them separately leads to $n = m$ and $\forall i, length(\varphi t_i) = length(\sigma t'_i) = 1$. According to Definition 3, we derive $\forall i, t_i = \sigma t'_i$ and $t'_i = \varphi t_i$. If $t_i$ is not a variable, $t'_i = \varphi t_i = t_i$, i.e., all non-variable tokens in $s_1$ and $s_2$ are identical. If $t_i$ is a variable, $t'_i$ must also be a variable because otherwise $t_i = \sigma t'_i$ would not be a variable. Therefore, both $\sigma$ and $\varphi$ are just renaming variables. And it is straightforward to verify that they cannot map different variables to the same. In other words, $\sigma$ and $\varphi$ are $\alpha-$conversions; $s_1 = s_2$ modulo $\alpha$-conversion.

3. Given three sentences $s_1$, $s_2$, and $s_3$ such that $s_1 \leq s_2$ and $s_2 \leq s_3$, there exist substitutions $\sigma$ and $\varphi$ such that $s_1 = \sigma s_2$ and $s_2 = \varphi s_3$. Let $\mu = \sigma \circ \varphi$ be the function composite of $\sigma$ and $\varphi$. $\mu$ is also a substitution and $s_1 = \mu s_3$. Therefore, $s_1 \leq s_3$.

$\square$

**Theorem 2** (Partial order among rules). *The $\leq$ in Definition 4 is a partial order among rules. In other words,*

1. *For any rule $r$, $r \leq r$.*

2. *For any two rules $r_1$ and $r_2$, if $r_1 \leq r_2$ and $r_2 \leq r_1$, then $r_1 = r_2$ modulo $\alpha$-conversion.*

3. *For any three rules $r_1$, $r_2$ and $r_3$, if $r_1 \leq r_2$ and $r_2 \leq r_3$, then $r_1 \leq r_3$.*

*Proof.* Similar to the proof of Theorem 1. $\square$

**Definition 8** (Strictly partial order among sentences/rules). *Let $s_1$ and $s_2$ be two sentences, $s_2$ is strictly more general than $s_1$ (denoted by $s_1 < s_2$) if and only if $s_1 \leq s_2$ and $s_1 \neq s_2$ modulo $\alpha$-conversion. Similarly, if $r_1$ and $r_2$ are rules, $r_1 < r_2$ if and only if $r_1 \leq r_2$ and $r_1 \neq r_2$ modulo $\alpha$-conversion.*

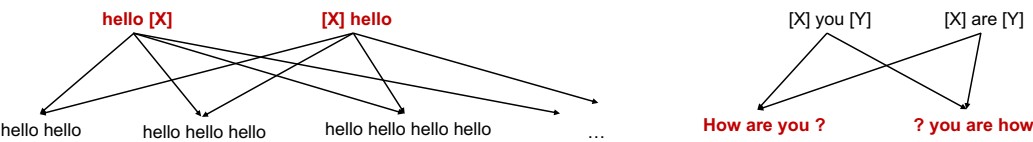

Figure 3: The minimal complete set of unifiers of two sentences can be empty, finite, or infinite (e.g., "`hello [X]`" and "`[X] hello`"). The minimal complete set of anti-unifiers is non-empty and finite.

## B  UNIFICATION AND ANTI-UNIFICATION OF SENTENCES AND RULES

Unification and anti-unification (Plotkin, 1970; Robinson & Voronkov, 2001) are basic symbolic procedures in formal logic that are useful for theorem proving and logic programming (Russell & Norvig, 2002; Yernaux & Vanhoof, 2019). In MetaQNL, unification is used in backward chaining, and anti-unification is used to abstract concrete rules into rules with variables. We adapt existing problem setups and algorithms from formal logic to MetaQNL. The algorithms we use for MetaQNL do not have theoretical guarantees as in formal logic, but they work well in practice.

**Unification.** Given two sentences (or two rules), unification aims to find substitutions mapping them to the same sentence (or rule). Such substitutions are called unifiers. We extend unification to MetaQNL by adapting prior work, especially the unification algorithm developed by Kutsia (2002) for a variant of first-order logic with sequence variables and flexible arity symbols.

**Definition 9** (Unifier). *A substitution $\sigma : \Sigma_v \to \Sigma_{-s}^+$ is a unifier of two sentences $s_1, s_2 \in \Sigma^+$ if and only if $\sigma s_1 = \sigma s_2$. Similarly, it is a unifier of two rules $r_1$ and $r_2$ if and only if $\sigma r_1 = \sigma r_2$.*

Two sentences may have multiple unifiers. Taking $s_1 = $ `[X] is [Y]`, $s_2 = $ `The elphant [Z]` as an example, their unifiers include $\sigma = \{$`[X]` $\to$ `The elephant,` `[Z]` $\to$ `is [Y]`$\}$, $\varphi = \{$`[X]` $\to$ `The elephant,` `[Y]` $\to$ `tall,` `[Z]` $\to$ `is tall`$\}$, etc. Both $\sigma$ and $\varphi$ are valid unifiers, but they lead to different sentences when applied: $\sigma s_1 = $ `The elephant is [Y]`, $\varphi s_1 = $ `The elephant is tall`. We prefer $\sigma$ to $\varphi$ because it is more general; it does not introduce any new information not in $s_1$ and $s_2$. In contrast, we cannot infer the "`tall`" in $\varphi$. This is the intuition behind the concept of "most general unifiers".

**Definition 10** (Most general unifier). *Let the substitution $\sigma$ be a unifier of sentencens $s_1$ and $s_2$, it is a most general unifier if and only if there is no unifier $\varphi$ of $s_1$ and $s_2$ such that $\sigma s_1 < \varphi s_1$.*

In unification, we want to compute a set of most general unifiers, and we want the set to be minimal and complete. Below we define these concepts for sentences.

**Definition 11** (Complete set of unifiers). *Let $\mathcal{U}$ be a set of unifiers of sentences $s_1$ and $s_2$, $\mathcal{U}$ is complete if and only if for any unifier $\varphi$ of $s_1$ and $s_2$, there exists a unifier $\sigma \in \mathcal{U}$, such that $\varphi s_1 \leq \sigma s_1$.*

**Definition 12** (Minimal set of unifiers). *Let $\mathcal{U}$ be a set of unifiers of sentences $s_1$ and $s_2$, $\mathcal{U}$ is minimal if and only if for any $\sigma, \varphi \in \mathcal{U}$, $\varphi s_1 \leq \sigma s_1$ implies $\sigma = \varphi$ (modulo $\alpha$-conversion).*

**Definition 13** (Minimal complete set of unifiers). *Let $\mathcal{U}$ be a set of unifiers of sentences $s_1$ and $s_2$, $\mathcal{U}$ is a minimal complete set of unifiers if and only if it is both minimal and complete.*

The definitions for rules are parallel. Given two sentences (or two rules), the unification problem is to compute a minimal complete set of unifiers. The result can be empty (e.g., unifying "`hello world`" and "`how are you`"), finite ("`hello [X]`" and "`[Y] world`"), or infinite ("`hello [X]`" and "`[X] hello`", Fig. 3 *Left*).

**Anti-unification.** Given two sentences (or two rules), anti-unification aims to generalize them into a single sentence (or rule). Anti-unification has also been studied in formal logic (Plotkin, 1970; Kutsia et al., 2014). We extend it to MetaQNL by adapting prior work. For simplicity, we define anti-unification only for sentences, but it applies to rules as well.

**Definition 14** (Anti-unifier). *Given two sentences $s_1$ and $s_2$, their anti-unifier is a triple $(s, \sigma_1, \sigma_2)$ of a sentence $s$ and two substitutions $\sigma_1, \sigma_2$, such that $\sigma_1 s = s_1$ and $\sigma_2 s = s_2$.*

**Definition 15** (Most specific anti-unifier). *Let $(s, \sigma_1, \sigma_2)$ be an anti-unifier of sentencens $s_1$ and $s_2$, it is a most specific anti-unifier if and only if there is no substitution $\varphi$, $\sigma_1'$ and $\sigma_2'$ such that*

1. *$\sigma_1 = \sigma_1' \circ \varphi$, $\sigma_2 = \sigma_2' \circ \varphi$*

2. *$\varphi s < s$*

3. *$(\varphi s, \sigma_1', \sigma_2')$ is also an anti-unifier of $s_1$ and $s_2$*

**Definition 16** (Complete set of anti-unifiers). *Let $\mathcal{A}$ be a set of anti-unifiers of sentences $s_1$ and $s_2$, $\mathcal{A}$ is complete if and only if for any anti-unifier $(s, \sigma_1, \sigma_2)$ of $s_1$ and $s_2$, there exists a substitution $\varphi$ and an anti-unifier $(\varphi s, \sigma_1', \sigma_2')$ such that $\sigma_1 = \sigma_1' \circ \varphi$, $\sigma_2 = \sigma_2' \circ \varphi$.*

**Definition 17** (Minimal set of anti-unifiers). *Let $\mathcal{A}$ be a set of anti-unifiers of sentences $s_1$ and $s_2$, $\mathcal{A}$ is minimal if and only if for any $(s, \sigma_1, \sigma_2), (s', \sigma_1', \sigma_2') \in \mathcal{A}$, if there exists a substitution $\varphi$ such that*

1. *$s' = \varphi s$*

2. *$\sigma_1 = \sigma_1' \circ \varphi$, $\sigma_2 = \sigma_2' \circ \varphi$*

*then $\varphi$ must be an $\alpha$-conversion, i.e. $\varphi s = s$ (modulo $\alpha$-conversion).*

**Definition 18** (Minimal complete set of anti-unifiers). *Let $\mathcal{A}$ be a set of anti-unifiers of sentences $s_1$ and $s_2$, $\mathcal{A}$ is a minimal complete set of anti-unifiers if and only if it is both minimal and complete.*

Given two sentences (or two rules), the anti-unification problem is to compute a minimal complete set of anti-unifiers. Unlike unification, the result of anti-unification must be non-empty and finite (Fig. 3).

**Theorem 3** (Anti-unification is finitary). *Let $\mathcal{A}$ be a minimal complete set of anti-unifiers of sentences $s_1$ and $s_2$, then $\mathcal{A}$ is non-empty and finite.*

*Proof.* For any sentences $s_1$ and $s_2$, we have a trivial anti-unifier $([X], \sigma_1, \sigma_2)$ where $\sigma_1 = \{[X] \to s_1\}$ and $\sigma_2 = \{[X] \to s_2\}$. Since $\mathcal{A}$ is complete, apply Definition 16 and we will know $\mathcal{A}$ must be non-empty.

For any anti-unifier $(s, \varphi_1, \varphi_2) \in \mathcal{A}$, we have $\varphi_1 s = s_1$ (Definition 14). Apply Lemma 1 to derive $\text{length}(s) \leq \text{length}(\varphi_1 s) = \text{length}(s_1)$. Therefore, the length of $s$ is bounded. Also, $s$ cannot have non-variable tokens besides those in $s_1$, so its vocabulary is also bounded. There are finite number of different sentences that $s$ can take (modulo $\alpha$-conversion). Therefore, $\mathcal{A}$ must also be finite. $\square$

Our current anti-unification algorithm is adapted from Kutsia et al. (2014). It recursively matches the beginning of two sentences. Let $s_1$ and $s_2$ be sentences, and $s$ is a more general sentence in their anti-unifier. If $s_1$ and $s_2$ start with the same word $w$, $s$ should also start with $w$. Otherwise, $s$ should start with a variable corresponding to some prefixes of $s_1$ and $s_2$. The algorithm searches for all such prefixes and anti-unifies the remaining parts of the sentences recursively.

## C  DETAILS OF MINISCAN EXPERIMENTS

The 14 MiniSCAN (Lake et al., 2019) training examples represented as sentences in MetaQNL ($MAPS\_TO$ is a special symbol):

```
                          dax  $MAPS_TO$  RED
                          lug  $MAPS_TO$  BLUE
                          wif  $MAPS_TO$  GREEN
                          zup  $MAPS_TO$  YELLOW
                      dax fep  $MAPS_TO$  RED RED RED
                      lug fep  $MAPS_TO$  BLUE BLUE BLUE
              wif blicket dax  $MAPS_TO$  GREEN RED GREEN
              lug blicket wif  $MAPS_TO$  BLUE GREEN BLUE
                 dax kiki lug  $MAPS_TO$  BLUE RED
                 lug kiki wif  $MAPS_TO$  GREEN BLUE
             lug fep kiki wif  $MAPS_TO$  GREEN BLUE BLUE BLUE
             lug kiki wif fep  $MAPS_TO$  GREEN GREEN GREEN BLUE
      wif kiki dax blicket lug  $MAPS_TO$  RED BLUE RED GREEN
      wif blicket dax kiki lug  $MAPS_TO$  BLUE GREEN RED GREEN
```

The 10 testing examples:

```
                      zup fep  $MAPS_TO$  YELLOW YELLOW YELLOW
              zup blicket lug  $MAPS_TO$  YELLOW BLUE YELLOW
                 zup kiki dax  $MAPS_TO$  RED YELLOW
             zup fep kiki lug  $MAPS_TO$  BLUE YELLOW YELLOW YELLOW
             wif kiki zup fep  $MAPS_TO$  YELLOW YELLOW YELLOW GREEN
       lug kiki wif blicket zup  $MAPS_TO$  GREEN YELLOW GREEN BLUE
  zup blicket wif kiki dax fep  $MAPS_TO$  RED RED RED YELLOW GREEN YELLOW
  zup blicket zup kiki zup fep  $MAPS_TO$  YELLOW YELLOW YELLOW YELLOW YELLOW YELLOW
             dax blicket zup  $MAPS_TO$  RED YELLOW RED
                 wif kiki zup  $MAPS_TO$  YELLOW GREEN
```

The rule proposer generates as candidates 1288 concrete rules with $\leq 2$ premises, by combining the 14 training sentences in all possible ways. Below are some examples. Note that most candidate rules are wrong.

```
                    lug fep $MAPS_TO$ BLUE BLUE BLUE  ⊢  dax $MAPS_TO$ RED
dax kiki lug $MAPS_TO$ BLUE RED; wif $MAPS_TO$ GREEN  ⊢  lug kiki wif $MAPS_TO$
                                                                  GREEN BLUE
  lug $MAPS_TO$ BLUE; dax fep $MAPS_TO$ RED RED RED  ⊢  dax $MAPS_TO$ RED
```

MetaInduce learns 7 rules corresponding to the ground truth rules of MiniSCAN:

```
                                      ⊢  dax $MAPS_TO$ RED
                                      ⊢  lug $MAPS_TO$ BLUE
                                      ⊢  wif $MAPS_TO$ GREEN
                                      ⊢  zup $MAPS_TO$ YELLOW
                  [A] $MAPS_TO$ [B]  ⊢  [A] fep $MAPS_TO$ [B] [B] [B]
[A] $MAPS_TO$ [B]; [C] $MAPS_TO$ [D]  ⊢  [A] kiki [C] $MAPS_TO$ [D] [B]
[A] $MAPS_TO$ [B]; [C] $MAPS_TO$ [D]  ⊢  [A] blicket [C] $MAPS_TO$ [B] [D] [B]
```

# D  DETAILS OF SCAN EXPERIMENTS

Some examples in SCAN (Lake & Baroni, 2018):

```
                  walk  $MAPS_TO$  WALK
                  jump  $MAPS_TO$  JUMP
            turn right  $MAPS_TO$  RIGHT
   jump after turn left  $MAPS_TO$  LEFT JUMP
            walk right  $MAPS_TO$  RIGHT WALK
         walk after run  $MAPS_TO$  RUN WALK
        turn left twice  $MAPS_TO$  LEFT LEFT
     turn opposite left  $MAPS_TO$  LEFT LEFT
     turn around right  $MAPS_TO$  RIGHT RIGHT RIGHT RIGHT
      walk around left  $MAPS_TO$  LEFT WALK LEFT WALK LEFT WALK LEFT WALK
```

The rule proposer is similar as before, but with an additional filtering heuristic based on compositionality (see Sec. 5). Below are some examples of the candidate rules it generates. Note that many of them are wrong because the premises are not sufficient to deduce the conclusion.

```
                        run $MAPS_TO$ RUN  ⊢  walk after run $MAPS_TO$ RUN WALK
walk $MAPS_TO$ WALK; run $MAPS_TO$ RUN  ⊢  walk after run $MAPS_TO$ RUN WALK
                        run $MAPS_TO$ RUN  ⊢  jump twice after run twice $MAPS_TO$
                                              RUN RUN JUMP JUMP
           run twice $MAPS_TO$ RUN RUN  ⊢  jump twice after run twice $MAPS_TO$
                                              RUN RUN JUMP JUMP
```

MetaInduce learns 20 rules corresponding to the ground truth rules of SCAN:

```
                                    ⊢  walk $MAPS_TO$ WALK
                                    ⊢  look $MAPS_TO$ LOOK
                                    ⊢  run $MAPS_TO$ RUN
                                    ⊢  jump $MAPS_TO$ JUMP
                                    ⊢  turn right $MAPS_TO$ RIGHT
                                    ⊢  turn left $MAPS_TO$ LEFT
                                    ⊢  turn opposite left $MAPS_TO$
                                          LEFT LEFT
                                    ⊢  turn opposite right $MAPS_TO$
                                          RIGHT RIGHT
                                    ⊢  turn around left $MAPS_TO$
                                          LEFT LEFT LEFT LEFT
                                    ⊢  turn around right $MAPS_TO$
                                          RIGHT RIGHT RIGHT RIGHT
              [A] $MAPS_TO$ [B]  ⊢  [A] left $MAPS_TO$ LEFT [B]
              [A] $MAPS_TO$ [B]  ⊢  [A] right $MAPS_TO$ RIGHT [B]
              [A] $MAPS_TO$ [B]  ⊢  [A] opposite left $MAPS_TO$
                                          LEFT LEFT [B]
              [A] $MAPS_TO$ [B]  ⊢  [A] opposite right $MAPS_TO$
                                          RIGHT RIGHT [B]
              [A] $MAPS_TO$ [B]  ⊢  [A] around left $MAPS_TO$
                                          LEFT [B] LEFT [B] LEFT [B] LEFT [B]
              [A] $MAPS_TO$ [B]  ⊢  [A] around right $MAPS_TO$
                                          RIGHT [B] RIGHT [B] RIGHT [B] RIGHT [B]
              [A] $MAPS_TO$ [B]  ⊢  [A] twice $MAPS_TO$ [B] [B]
              [A] $MAPS_TO$ [B]  ⊢  [A] thrice $MAPS_TO$ [B] [B] [B]
[A] $MAPS_TO$ [B]; [C] $MAPS_TO$ [D]  ⊢  [C] and [A] $MAPS_TO$ [D] [B]
[A] $MAPS_TO$ [B]; [C] $MAPS_TO$ [D]  ⊢  [A] after [C] $MAPS_TO$ [D] [B]
```

## E    DETAILS OF RULETAKER EXPERIMENTS

**Rule proposer.**

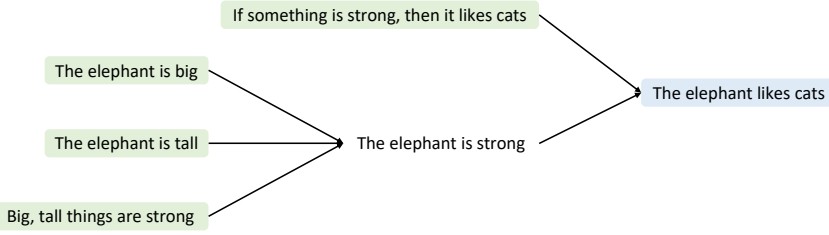

Figure 4: RuleTaker contains ground truth proofs in the form of directed acyclic graphs from the assumptions to the conclusion. The nodes in the graph are concrete sentences without variables.

Fig. 4 shows the form of ground truth proofs in RuleTaker. For this specific example, our rule proposer would generate 2 candidate rules below:

```
                      $TRUE$ The elephant is big;
                      $TRUE$ The elephant is tall;
                      $TRUE$ Big , tall things are strong;
                  ⊢  $TRUE$ The elephant is strong

                $TRUE$ The elephant is strong;
                $TRUE$ If something is strong , then it likes cats;
              ⊢  $TRUE$ The elephant likes cats
```

**Learned rules.** Below are some example rules learned on RuleTaker:

```
        $TRUE$ the [A] likes the [B];
        $TRUE$ the [A] is [C];
        $TRUE$ if someone is [C] and they like the [B] then the [D];
      ⊢  $TRUE$ the [D]

                  $TRUE$ the [A] does not see the [B];
                ⊢  $FALSE$ the [A] sees the [B]

        $TRUE$ the [A] [B] the [C];
        $TRUE$ the [C] [D];
        $TRUE$ if someone [B] the [C] and the [C] [D] then they need the [E];
      ⊢  $TRUE$ the [A] needs the [E]

                  $TRUE$ [A] is [B];
                  $TRUE$ [A] is [C];
                  $TRUE$ [B] , [C] things are [E];
                ⊢  $TRUE$ [A] is [E]

        $TRUE$ the [A] is [B];
        $TRUE$ the [A] is [C];
        $TRUE$ if someone is [C] and [B] then they chase the [D];
      ⊢  $TRUE$ the [A] chases the [D]

                  $TRUE$ [A] is [B];
                ⊢  $FALSE$ [A] is not [B]
```

## F  HEURISTICS FOR CONSTRAINING THE SPACE OF RULES

We use a few simple and general heuristics for constraining the space of rules and pruning invalid rules generated by anti-unification.

First, we merge multiple variables that always appear together. For example, the `[A]` `[B]` `[C]` and `[D]` `[E]` in the rule below can be merged.

```
      $TRUE$ If [A] [B] [C] then [D] [E];
      $TRUE$ [A] [B] [C];
⊢  $TRUE$ [D] [E]
```

So the rule becomes:

```
      $TRUE$ If [A] then [B];
      $TRUE$ [A];
⊢  $TRUE$ [B]
```

A variable in a rule is called a *free variable*, if it appears only once. For example, the rule

```
   If something is red, then tomorrow will be sunny;
   [X] is red;
⊢  Tomorrow will be sunny
```

contains a free variable `[X]`. We require that a rule cannot contain free variables in its conclusion. Because they would allow arbitrary conclusions formed by substituting them with other sentences. For example, the rule below is not allowed because of the free variable `[X]` in the conclusion:

```
   $TRUE$ Today is sunny;
⊢  $TRUE$ Tommorow is [X]
```

In addition, a rule cannot contain a premise made of one single free variable. Because this premise can be satisfied by any sentence, and there is no point including it in the rule. For example, the rule below is not allowed because of the free variable `[X]`:

```
      $TRUE$ [X];
      $TRUE$ If [A] then [B];
      $TRUE$ [A];
⊢  $TRUE$ [B]
```

Finally, for RuleTaker, we only consider rules with no more than 1 free variable.

