# OpenReview forum: "Learning Symbolic Rules for Reasoning in Quasi-Natural Language"
_ICLR.cc/2022/Conference — ICLR 2022 Submitted_

### Official Review · Reviewer_7F3Z · 2021-10-25

**Correctness:** 4
**Technical Novelty And Significance:** 3
**Empirical Novelty And Significance:** 3
**Recommendation:** 8
**Confidence:** 3

**Main Review:**

Overall, I liked the scope of this paper, the importance, and non-triviality of the problem, and the novelty of the proposed approach. In my view, it certainly adds a dimension to the literature on symbolic rule learning. The idea of MetaQNL is simple yet effective to handle the natural language inputs within a formal symbolic system. The idea behind MetaInduce is also quite natural. Although there are some weaknesses of the proposed approach, I still feel this is a novel idea and has the potential to yield something big in the future.

__Strength__

- A well-written paper.
- A very nice literature survey and positioning of the work relative to the prior art.
- An important problem in the broad space of AI.

__Weakness__

- Time complexity of each of the three steps in MetaInduce is not discussed. It will be good to shed some light on this.
- As stated in the limitation section, the proposed approach is far from mature but serves as proof of concept.  It does not scale to millions of training examples.

**Summary Of The Paper:**

Traditional research on symbolic reasoning assumes input data are already translated into a format that complies with the formalism of the underlying system. A major struggle for symbolic reasoning is to handle the data coming in a natural form (e.g. images/text) and perform reasoning on the same. Inspired by this challenge, this paper undertakes the problem of converting text inputs into the format of a system formalism.

The formalism is also proposed by this paper and is called MetaQNL. The framework of MetaQNL is designed keeping in mind the specific need of operating directly on natural language sentences. This formalism supports the natural language sentences with variables within them. This trick alleviates the need of using a semantic parser to parse natural language sentences for the purpose of logical reasoning. Thus, MetaQNL, by design, is conducive to working with natural language inputs.

Next, this paper proposes an algorithm to induce rules from natural language inputs within the MetaQNL framework. This paper doesn’t worry about performing actual deductive reasoning/theorem proving and instead proposes to use existing provers and instead focus on the more challenging problem of rule induction. MetaInduce algorithm draws inspiration from existing ILP approaches. MetaInduce encodes the rule induction problem as a maximum satisfiability (MAX-SAT) problem, which can be solved efficiently by off-the-shelf solvers. The proposed method consists of 3 steps.

1. Given a training example, a rule proposer proposes a set of concrete rules as candidates. This set can be overcomplete/inaccurate.
2. It generates abstract rules from concrete rules via a symbolic procedure called anti-unification. This is essentially a process of aligning common substring segments across two or more strings.
3. It encodes the proof paths in MAX-SAT and solves for a subset of all rules using a MAX-SAT solver.

This paper benchmarks the proposed method on 2 tasks - learning compositional instructions and logical reasoning. For learning compositional instructions, it works on two standard benchmarks: MiniSCAN and SCAN and recovers precisely the ground truth rules. For logical reasoning, the proposed method achieves SOTA on the RuleTaker dataset.



**Summary Of The Review:**

See my comments in the main review.

---

> ### Author Response · Authors · 2021-11-19
> **Individual Response to Reviewer 7F3Z**
>
> Thank you for your valuable feedback! In the common response above, we address your questions about the time complexity of MetaInduce and how to potentially scale it to more training examples. Please feel free to post additional comments if you have further questions.

---

> > ### Comment · Reviewer_7F3Z · 2021-11-29
> > **Thoughts after reading authors' feedback**
> >
> > Thank you for addressing my concern regarding time complexity. I guess your answer is fair given the hardness and nature of the problem. I have no more questions.

---

### Official Review · Reviewer_9uxa · 2021-10-30

**Correctness:** 3
**Technical Novelty And Significance:** 4
**Empirical Novelty And Significance:** 2
**Recommendation:** 6
**Confidence:** 5

**Main Review:**

Overall the idea is novel and I like the paper presentation. However, I have some concerns regarding its ability to work with real natural language and the reproducibility of the proposed approach.

MAJOR CONCERN 1:
I have a concern that the rule proposer and the anti-unification require very well-formed of the sentences to be working well.

As we can see with the example on page 7 figure (a) about anti-unification, the sample shows very simple rules. I wonder how it will work when the sentence is really complicated with real natural language? Also the experiments in RuleTaker only with synthetically generated sentences in a very controlled language, could you please  demonstrate your results with complicated sentences beyond synthetic texts, I think the paraphrased datasets within the RuleTakers even not really natural language but that could be a good exercise for your methods to test on.


MAJOR CONCERN 2:
I worry about reproducibility as the source code is not open but the details explanation of the proposed approaches are missing. For example, from the paper I don't know how the rule proposer work and how the anti-unification is implemented with quasi-natural language. I would suggest to provide very detailed about the methods, with the current information I doubt that people can reimplement your work and reproduce what you have demonstrated.

MAJOR CONERN 3:
In the RuleTaker example, it is known that Transformers are good at generalisation when they are trained on queries with depth 3 or greater. Yet transformers are not good at generalisation when it is trained on lower depth queries. Could you please provide comparison results with Transformers when it is trained at lower depths?
Other minor comments:


"In contrast, our approach does not require a semantic parser, because rules in MetaQNL are directly applicable to natural language." ---> This is a strong statement, it requires a support with real natural language examples rather than synthetically generated sentences in the experiments.


Definition 6: what happen if there is no proof for a goal and the goal is proved via the close world assumption?



**Summary Of The Paper:**

The work proposes two new concepts:
+ MetaQNL: a symbolic system in Quasi-Natural Language. Instead of representing rules in a formal symbolic format such as first order logic rules, in MetaQNL, a rule is represented in a Quasi-Natural Language format which includes words, variables and control symbols. Since the rules are represented in an informal representation. An interesting property of the MetaQNL representation is that it allows to perform backward or forward inference by substitution of variables with sentences. The authors assume that texts can be translated into the MetaQNL format and thus solving the reasoning problems with text input is possible via mining rules from text and backward/forward reasoning with the Quasi-Natural Language.
+ To mine rule from text, the authors proposes an algorithm called MetaInduce. MetaInduce iterates through the training data several time to build a compact set of rules that trades complexity to prediction accuracy. It is a bottom up rule induction approach where it includes a Rule proposer which propose a concrete rule from a training example and an anti-unification module to abstract the concrete rule with more generalised rules. A pruning process based on MAX-SAT is used to prune the set of rules such that it optimise the regularised objective.

**Summary Of The Review:**

The paper proposes a new concept called quasi-language which allows representing rules in an new informal format that still allows to perform forward or backward reasoning while it is assumed to be mined easily from texts. The experimental results with some datasets with synthetically generated texts show that the methods work very well and advance state-of-the-art results.
However, I doubt the application of the work with natural language input, I explicitly request to perform more experiments with paraphrased datasets in RuleTaker (MAJOR CONCERN 1). I also have a concern about reproducibility as the presentation lacks details of the core components of the proposed algorithm (MAJOR CONCERN 2). I also requested an additional experiment regarding the training data with low depth queries to check the ability of generalization of the proposed approaches.

---

> ### Author Response · Authors · 2021-11-19
> **Individual Response to Reviewer 9uxa**
>
> Thank you for your valuable feedback! Below we address your questions and concerns. Please feel free to post additional comments if you have further questions.
>
>
> ## Experiments on real-world non-synthetic data
>
> Please see the common response above.
>
>
> ## Reproducibility.
>
> We will release the code for reproducing our experiments.
>
>
> ## Comparison with ProofWriter trained on lower-depth proofs.
>
> We conducted additional experiments of our method trained on the D1 part of RuleTaker (proof depth ≤ 1) and tested on D5. Results are shown in the table below (similar to Table 2 in the paper).
>
>
> | Test proof depth | N/A | 0        | 1        | 2       | 3      | 4      | 5      | All  |
> | ----------------- | ----- |------- | ------ | ----- | ------ | ----- | ----- | ---- |
> | Accuracy          | 99.3 | 100.0 | 100.0 | 99.8 | 98.6 | 98.4 | 98.3 | 99.2 |
>
> Our results when training on D1 are very close to when training on D3, showing that our method can generalize to longer proofs unseen in training. We are unable to compare with ProofWriter directly under this setting, because they haven’t released code.
>
>
> ## What if there is no proof for a goal and the goal is proved via the close world assumption?
>
> MetaQNL assumes the open-world assumption. Currently, it doesn't have a mechanism to support the negation as failure inference in the closed-world assumption. It is interesting to extend MetaQNL to the closed-world assumption. However, we believe the open-world assumption is a more realistic setting for accommodating incomplete information—in most real-world scenarios, there are certainly many statements that we don’t have enough information to either prove or disprove.

---

> > ### Comment · Reviewer_9uxa · 2021-11-19
> > **On additional experiments on natural language**
> >
> > Thank you for the team's hard work to respond to my reviews!
> >
> > Could you please kindly explain why you did not perform experiments on the paraphrased RuleTaker datasets as I requested but chose another dataset to do experiments? I think the paraphrased dataset of RuleTaker is a good exercise for the proposed approach because it will show how robust your approach w.r.t. to small changes in synthetic language via paraphrasing.

---

> > > ### Author Response · Authors · 2021-11-21
> > > **Response to the Additional Question from Reviewer 9uxa**
> > >
> > > We agree that the paraphrased RuleTaker dataset is a great suggestion! We only reported the morphology results because we had started working on it before the reviews were released. Therefore we were able to present relatively complete results despite the tight time frame.
> > >
> > > As for paraphrased RuleTaker, we have been actively working on it upon receiving the suggestion. Our initial results show that our system did not work well when applied directly: with linguistic variations, our system learns overly general rules. We believe the main reason is that the training set is too small (28K) for learning invariance to linguistic variations (many different ways of expressing the same thing), and we believe that training from such a small training set from scratch will be challenging for not just our method but also other learning-based methods. ProofWriter finetunes a T5-11B model pretrained on the C4 dataset with hundreds of millions of examples. The pretraining data exposes the model to abundant examples of linguistic variations, which may not be learnable from paraphrased RuleTaker alone. It would be interesting to see if ProofWriter can work without pretraining, but we are unable to perform this comparison since the code of ProofWriter is not publicly available. However, many works on large language models (e.g., [I], [J], [K]) reported performance drop when the pretraining data is downsized. The smallest datasets explored by those papers were still orders of magnitude larger than paraphrased RuleTaker.
> > >
> > > We believe that soft matching, which is a simple extension we have shown to be useful for the morphology task, is the key to addressing linguistic variations for paraphrased RuleTaker. We could use a pretrained language model to output matching scores between rules and assumptions. Soft matching allows outputting an approximate proof when a rigorous proof is not possible, with a score indicating the degree of rigor. We are investigating this extension as future work.
> > >
> > >
> > > * [I] Popel, Martin, and Ondřej Bojar. "Training tips for the transformer model." arXiv preprint arXiv:1804.00247 (2018).
> > > * [J] Kaplan, Jared, et al. "Scaling laws for neural language models." arXiv preprint arXiv:2001.08361 (2020).
> > > * [K] Raffel, Colin, et al. "Exploring the Limits of Transfer Learning with a Unified Text-to-Text Transformer." Journal of Machine Learning Research 21.140 (2020): 1-67.

---

### Official Review · Reviewer_CPsz · 2021-11-03

**Correctness:** 4
**Technical Novelty And Significance:** 3
**Empirical Novelty And Significance:** 2
**Recommendation:** 5
**Confidence:** 3

**Main Review:**

Pros:
- The learning algorithm is interesting and the problem of automatically discovering prepositions from examples is important for ATP in formal or informal languages.
- The paper clearly defines the terms used and explains the methods and experiments well.

Cons:
- As the authors pointed out, the experiments are neither large-scale nor real-world. One result of this is existing methods already achieve good performance, and it's not clear that the proposed method results in better performance.
- None of the components (theorem prover, rule abstraction, rule pruning) are novel individually.

Questions:
1) Are TRUE FALSE and MAPS_TO the only special symbols? It would be helpful to state this.
2) Are there any unprovable examples in SCAN?
3) The number of rules and symbols learned by MetaInduce is hard to compare with the number of learned parameters in ProofWriter. Is there a better metric to compare the two methods on?
4) What are the advantages to using the proposed symbol system instead of first order logic?

**Summary Of The Paper:**

This paper proposes an algorithm that learns rules from natural language data, and a symbolic system for manipulating these rules, where existing provers can be applied. The objective is to maximize the number of examples in a test set that are consistent with the proposed mode while minimizing the number of rules in the model. The algorithm consists of three steps - given a training example, it proposes concrete rules, abstracts concrete rules into rules with variables, and prunes the resulting rules.

**Summary Of The Review:**

The paper proposes an interesting solution to an important problem, but as-is the experimental settings and results are not compelling.

---

> ### Author Response · Authors · 2021-11-19
> **Individual Response to Reviewer CPsz**
>
> Thank you for your valuable feedback! Below we address your questions and concerns. Please feel free to post additional comments if you have further questions.
>
>
> ## Experiments on real-world data
>
> Please see the common response above.
>
>
> ## MetaQNL/MetaInduce do not significantly outperform deep neural networks
>
> Our method is a novel learning paradigm that radically differs from predominant methods based on deep neural networks, e.g., it learns symbolic rules without any continuous weights. Our experiments have demonstrated its promise in learning more compact and interpretable models, but yes, it does not significantly outperform deep neural networks right away. As we have discussed in Sec. 6, substantial future developments beyond a single paper are needed.
>
>
> ## MetaInduce consists of existing components for theorem proving, rule abstraction, and rule pruning.
>
> Yes, MetaInduce uses existing components, but this does not diminish  the novelty of our method. First, the MetaQNL symbolic system is our novel contribution. MetaInduce is proposed as an effective framework for learning MetaQNL rules from data. And its novelty cannot be disentangled with the novelty of MetaQNL. Second, MetaInduce itself is not a simple combination of existing components. We have discussed its difference with existing ILP methods in paragraphs 4–6 on page 4. And we elaborate more below:
>
> MetaInduce is the most similar to meta-level inductive logic programming (ILP) [B, C, D, E, F] in that it delegates rule induction to existing solvers (SAT, MAX-SAT, ASP, etc.). However, there are a few differences:
>
> 1. Encoding rules vs. encoding proofs: Most meta-level ILP approaches (e.g., ASPAL [B], ILASP [D], Apperception [E], and Popper [F]) directly encode candidate rules in answer set programming (ASP) and ask the ASP solver to find a subset of them, without a separate theorem proving stage. This is possible because they learn rules in Prolog, Datalog, or ASP, whose formal semantics can be encoded in ASP. In contrast, MetaQNL rules cannot be encoded in existing solvers directly. Therefore, we encode the proofs found by a prover.
>
> 2. Encoding proofs has been explored in ProSynth [C] for the provenance-guided synthesis of Datalog programs. However, we extend the SAT encoding of ProSynth. First, we encode the disjunction of all proof paths, whereas only one proof path is available in ProSynth due to how provenance works. Second, we have additional constraints about rule instantiation. Third, we can tolerate noise, whereas ProSynth cannot. We use soft constraints to enforce the training examples to be provable, but ProSynth uses hard constraints.
>
> * [B] Corapi et al.. "Inductive logic programming in answer set programming." International conference on inductive logic programming. 2011
> * [C] Raghothaman et al. "Provenance-guided synthesis of Datalog programs." POPL. 2019
> * [D] Law et al. "Inductive learning of answer set programs." European Workshop on Logics in Artificial Intelligence. 2014
> * [E] Evans et al. "Making sense of sensory input." Artificial Intelligence 2021
> * [F] Cropper and Morel. "Learning programs by learning from failures." Machine Learning. 2021
>
>
> ## Are `$TRUE$`, `$FALSE$`, and `$MAPS_TO$` the only special symbols?
>
> These are the only special symbols in our experiments on MiniSCAN, SCAN, and RuleTaker. However, MetaQNL does not restrict what special symbols can be used. When instantiating MetaQNL on a new task/dataset, one is free to use additional special symbols as needed. For example, in our additional experiment on morphological analysis (see the common response above), we introduce special symbols `$LEMMA$` and `$TAG$`.
>
>
> ## Are there unprovable examples in SCAN?
>
> No, as explained in the 3rd paragraph of page 8.
>
>
> ## How to properly compare the model size with ProofWriter?
>
> We are not aware of any widely accepted metric for comparing the size of symbolic rules and continuous weights. Fortunately, it doesn’t take a precise  comparison to see our model (~2869 symbols) is much smaller than ProofWriter (> 11 billion parameters in T5-11B). One can browse our model using a text editor on a laptop, but one cannot even load a [T5-11B](https://huggingface.co/t5-11b) model using a GPU with 40GB memory.
>
>
> ## What is the advantage of MetaQNL compared to first-order logic?
>
> The paper has discussed the advantage of MetaQNL compared to purely symbolic methods such as first-order logic (page 1 and page 3). In summary:
> * Symbolic methods are not directly applicable to domains not amenable to rigid formalizations, such as commonsense reasoning and natural language.
> * To handle natural language, one could apply semantic parsing before symbolic methods. However, it requires (1) high-quality semantic parsers covering a wide range of texts; (2) an ontology of objects and predicates in the world. None of which can be achieved easily.
> * The syntax of MetaQNL is compatible with natural language sentences without semantic parsing.

---

> > ### Comment · Reviewer_CPsz · 2021-12-07
> > **Thank you for the update**
> >
> > Thank you for doing your hard work on responding the reviews. I appreciate the new experiments, but am still concerned that the method would not scale well to real problems in effectiveness or time complexity, but I adjusted the score based on the novelty of your approach.

---

### Official Review · Reviewer_v7TB · 2021-11-04

**Correctness:** 2
**Technical Novelty And Significance:** 2
**Empirical Novelty And Significance:** 2
**Recommendation:** 3
**Confidence:** 1

**Main Review:**

strengths:
1) this is indeed novel research problem - using a learning system together with a maxsat solver to make logical inference and identify rules.  There is a spectrum where on one end logical inference instances fully in natural language, and on the other end they are fully symbolic, and this paper falls somewhere inbetween.
2) The paper is generally well-written, and mathematical part of the paper seems to be correct.

Weakness:

I fail to see the real contributions in this paper.  The only novelty seems to me is MetaQNL and MetaInduce can solve formal systems as well as systems represented in quasi natural languages.  However, are there actual applications that can potentially benefit from this problem formulation?
One of the major claims of this paper is it can produces checkable proofs, but isn't Proofwriter also capable of generating proofs which seemed more impressive because it is a fully neural system without any explicit encodings of rules and has the potential of working with real languages.  I get that proofwriter has billions of parameters and MetaInduce learns a system with much fewer parameters, but then what is the difference between this and combinatorial optimization?

Furthermore, I failed to see a way to scale up this method.  In the end, it depends on a maxsat solver, which will become intractable quickly when there are more rules.

In summary, if the authors can present a real-world application that can potentially benefit from their system while other learning systems fail to do so and evaluate their method on a small real-world dataset, I would be less concerned.

Small comments and questions:
1) Would appreciate a few citations to support the claim "At a glance, this may appear a large departure from the conventional wisdom that learning-based systems, particularly deep networks, are far superior to rule-based systems, as history has demonstrated repeatedly."  The authors need to be specific what on tasks.  For example, NNs are far behind SAT solvers on propositional formulas (GQSAT as an example for learning-based system).
2) I fail to understand why rule proposers need to generate concrete rules, aren't all the rules already included in MetaQNL systems?

**Summary Of The Paper:**

This paper proposes a symbolic system in Quasi-Natural Language, MetaQNL, which is compatible with both logical inference and quasi natural language expressions, and where the basic building blocks are sentences and rules. The authors also propose MetaInduce, which learns to generalize a set of rules that explains the examples in MetaInduce.

MetaInduce consists of three mains steps:
1) a rule proposer proposes a set of concrete rules as candidates for each individual example. This set may not be fully correct and may not be the minimal explanations, and they are used to prove the example using forward/backward chaining.
2) the authors apply a symbolic procedure called anti-unification to generate abstract rules from concrete rules.
3) and finally, proof paths are encoded in MAX-SAT and a subset of all rules are solved using a MAX-SAT solver to find minimal possible explanations

This paper evaluates its methods on two synthetic datasets, and the authors claim to learn compact models with much less data, and produces answers as well as proofs

**Summary Of The Review:**

In short, while I acknowledge this work is novel, its setting is not very practical.  This paper would benefit from presenting a potential real-world application, or else some theoretical generalization results on how well their systems can learn.

---

> ### Author Response · Authors · 2021-11-19
> **Individual Response to Reviewer v7TB**
>
> Thank you for your valuable feedback! Below we address your questions and concerns. Please feel free to post additional comments if you have further questions.
>
>
> ## Potential real-world applications
>
> Please see the common response above.
>
>
> ## Doesn't ProofWriter generate checkable proofs?
>
> No, as explained in the 2nd paragraph of page 4, the proofs generated by ProofWriters cannot be checked mechanically by computers.
>
> This is not to be confused with whether the proof can be verified by humans. Take the proof in Fig. 1 of the ProofWriter paper (Tafjord et al. 2020) as an example; humans can verify it is a correct proof. But it is non-trivial to write a general computer program for checking all such proofs mechanically. In contrast, proofs in MetaQNL are mechanically checkable by checking the conditions in Definition 5 (Sec. 3).
>
>
> ## Difference between MetaInduce and combinatorial optimization.
>
> These two are not directly comparable.  MetaInduce uses combinatorial optimization (MAX-SAT specifically) to learn symbolic rules. This is analogous to deep learning using gradient descent to learn the weights of neural networks. MetaInduce and combinatorial optimization are not directly comparable for the same reason deep learning and gradient descent are not directly comparable.
>
>
> ## Why do rule proposers need to generate concrete rules, aren't all the rules already included in MetaQNL?
>
> No, rules are not included in MetaQNL. MetaQNL only defines the syntax of rules and how they can be used in theorem proving (Sec. 3). But it does not come with any built-in rules. Instead, rules are learned from data using MetaInduce.
>
>
> The learned rules take very different forms depending on the application domain. For example, a rule learned from SCAN might look like:
> ```
> [A] $MAPS_TO$ [B]
> —
> [A] twice $MAPS_TO$ [B] [B]
> ```
> , whereas a rule learned from RuleTaker might look like:
> ```
> If [A] then [B]
> [A]
> —
> [B]
> ```
> Therefore, instead of building domain-agnostic rules into MetaQNL, we use rule proposers to generate concrete rules for each domain.
>
>
> ## MAX-SAT solvers are not scalable.
>
> Please see the common response above.
>
>
> ## Additional citations.
>
> Thank you for the suggestion. We'll incorporate them in the next revision.

---

> > ### Comment · Reviewer_v7TB · 2021-11-30
> > **Thanks for your rebuttal**
> >
> > Thanks for doing so much work on responding my reviews.  While the new datasets are less toy problems, I'm still concerned about the practical use of these algorithms.  I also feel that the related literature is not solid.  For these reasons, I will keep my original score, but I have lowered my confidence in case I miss anything important in this paper.

---

### Author Response · Authors · 2021-11-19
**Common Response (1/3)**

We thank all reviewers for their thoughtful comments. We are encouraged that they agree we address an important problem (CPsz, 7F3Z). Reviewers consider our approach novel (v7TB, 9uxa, 7F3Z), interesting (CPsz), natural, simple yet effective (7F3Z), and our paper well-written (v7TB, CPsz, 9uxa, 7F3Z). We are especially pleased that reviewer 7F3Z thinks our work adds a dimension to the literature and has the potential to yield to something big in the future.

## Additional Experiments on Real-world Non-synthetic Data

A common concern among reviewers (v7TB, CPsz, 9uxa) is that our original experiments were synthetic and could not demonstrate how well the method works on real-world natural language. This is a legitimate concern. Due to the complex nature of natural language, extending our method to natural language may require non-trivial future developments beyond a single paper (also discussed in Sec. 6).

That said, we have conducted additional experiments on a real-world, non-synthetic task—analyzing the morphology of words. We hope the new experiments can mitigate the reviewers’ concerns and provide evidence that our current system can work with noisy real-world data to some extent.



 ## Task and dataset

We use the same task and dataset in Sec. 5.2 of Akyürek et al. [A]: Given the surface form of a word (e.g., `studied`), the model predicts its lemma (`study`) and an unknown number of morphological tags, such as`V` (verb), `SG` (singular), and `PST` (past tense).

The data is constructed from the SIGMORPHON 2018 dataset. It consists of 3 languages with varying morphological complexity—Spanish, Swahili, and Turkish. For each language, they sample a training set of 1000 examples and three test sets (FUT, PST, and OTHER) of 100 examples each. FUT consists exclusively of words in the future tense; PST consists of words in the past tense; OTHER consists of other words. The training set has only 8 past-tense words and 8 future-tense words. Therefore, FUT and PST test models' few-shot learning capabilities.

Although this task differs from general natural language understanding, it is still challenging. The morphological data is not synthetic; it contains noise and ambiguity, just like most real-world data. A standard seq2seq neural network performs far from perfect, especially on FUT and PST (E.g., an F1 score of 66% on Spanish; see Table 2 of Akyürek et al.).

* [A] Akyürek, Ekin, Afra Feyza Akyürek, and Jacob Andreas. "Learning to Recombine and Resample Data For Compositional Generalization." ICLR 2020.


## Method

### Data preprocessing

To apply our method to the task of morphological analysis, we represent both the surface form and the lemma as characters. The surface form serves as the assumption, whereas the lemma and the tags serve as conclusions. For example, below is a training example in the dataset:
```
Input surface form:  zarandeamos
Output lemma:        zarandear
Output tags:         V;IND;PRS;1;PL
```
We treat `z a r a n d e a m o s` as the assumption. Provable conclusions include `$LEMMA$ z a r a n d e a r`, `$TAG$ V`, `$TAG$ IND`, `$TAG$ PRS`, `$TAG$ 1`, and `$TAG$ PL`. Everything else is unprovable.
### Rule proposal

The rule proposer simply generates concrete rules that can prove the conclusions in a single step:
```
z a r a n d e a m o s
—
$LEMMA$ z a r a n d e a r
```

```
z a r a n d e a m o s
—
$TAG$ V
```

```
z a r a n d e a m o s
—
$TAG$ IND
```

```
z a r a n d e a m o s
—
$TAG$ PRS
```

```
z a r a n d e a m o s
—
$TAG$ 1
```

```
z a r a n d e a m o s
—
$TAG$ PL
```

---

> ### Author Response · Authors · 2021-11-19
> **Common Response (2/3)**
>
> ## Results
>
> We follow the same baseline and evaluation setup in Table 2 of Akyürek et al. The predicted lemmas and tags are evaluated using F1 score. The results on the FUT set and the PST set are averaged. The baseline is a standard seq2seq neural network: LSTMs with attention. The table below shows the results:
>
> | Model                          | Spanish FUT+PST | Spanish OTHER | Swahili FUT+PST | Swahili OTHER | Turkish FUT+PST | Turkish OTHER |
> | ------------------------- | --------------------- | ------------------- | -------------------- | ----------------- | -------------------- |  ----------------- |
> | LSTMs + attention       | **66**                     | **88**                   |  75                       | **90**                | **69**                    | **85**                 |
> | Ours                             | 55                          | 82                        | **81**                   | 86                     | 53                         | 71                      |
>
> Note that we’re comparing with the baseline in Akyürek et al., not their proposed method. Their method focuses on generating synthetic data for augmenting the training set. Therefore it is orthogonal to our work.
>
> And here are some example rules learned by our method on Spanish:
> ```
> [A] e a m o s
> —
> $LEMMA$ [A] e a r
> ```
>
> ```
> [A] á r a m o s
> —
> $TAG$ PST
> ```
>
> The results show that our method is competitive with the baseline on Swahili, but there are still gaps on Spanish and Turkish. By analyzing the model's predictions, we find different reasons for the gaps. Turkish morphology is known to be very complex. But our current way of instantiating MetaQNL only considers proofs of depth 1, which could be a restriction for learning more expressive rules.
>
> On the other hand, Spanish morphology is relatively simple. But the F1 score of our system is not great, because it learns a set of over-specific rules that achieve high precision but low recall. Next, we explore how to mitigate this issue through soft matching.
>
> ## Soft matching
>
> In the original MetaQNL, a rule is applicable only if its premises match the assumptions precisely. For example, the rule
> ```
> e [A] á r a m o s
> —
> $LEMMA$ e [A] a r
> ```
> is not applicable to `m u t i l á r a m o s` due to the mismatch between `e` and `m`.
>
> The rigid and precise matching could be a restriction if we want to apply the learned rules to noisy testing examples. One potential solution is to perform “soft matching”—applying rules without requiring precise matching. There are potentially many ways to implement soft matching, some of which may also leverage machine learning. Here we use a simple implementation based on anti-unification as a first step to explore whether soft matching could be useful.
>
> Given a rule such as
> ```
> e [A] á r a m o s
> —
> $LEMMA$ e [A] a r
> ```
> and an assumption such as `m u t i l á r a m o s`. The rule is not applicable, but we anti-unify `e [A] á r a m o s` and `m u t i l á r a m o s` to find a more general rule that is applicable:
> ```
> [A] á r a m o s
> —
> $LEMMA$ [A] a r
> ```
> All such additional rules are ranked based on how different they are from the original rules. Note that we only perform soft matching in testing, and the training process remains the same.
>
> Our preliminary results show that even this simple form of soft matching can bridge the performance gap on Spanish:
>
> | Model                          | Spanish FUT+PST | Spanish OTHER |
> | ------------------------- | --------------------- | ------------------- |
> | LSTMs + attention       | **66**                     | **88**                  |
> | Ours                             | 55                          | 82                       |
> | Ours + soft matching   | **66**                     |  84                      |
>
> However, it leads to no improvements on Swahili and Turkish. We did some analysis and found that the rules learned on Swahili and Turkish are more noisy, due to the increased morphological complexity. Therefore, relaxing the matching conditions naively may lead to too many spurious rules. In the future, it is interesting to explore more principled mechanisms for soft matching, which can potentially improve MetaQNL on noisy real-world domains.
>
>
> ## Limitations and Future work
>
> In summary, our experiments on non-synthetic data are a step towards applying MetaQNL to real-world problems that are complex and noisy. They show encouraging results (e.g., on Swahili) and reveal some remaining challenges (e.g., on Turkish). And they also suggest soft matching could be an important direction to explore in the future.
>
> We hope these experiments can mitigate the reviewers’ concerns and provide a more holistic picture of our method. However, we do not claim these experiments alone can already show MetaQNL to work on unconstrained natural language. Most challenges discussed in Sec. 6 still apply, but we believe resolving them requires efforts beyond a single paper.

---

> > ### Author Response · Authors · 2021-11-19
> > **Common Response (3/3)**
> >
> > ## Scalability of MetaInduce
> >
> > Reviewer v7TB and 7F3Z asked about how to potentially scale MetaInduce to more training examples. We approach this problem from two angles: theoretical and experimental.
> >
> > ### Time complexity of MetaInduce
> >
> > MetaInduce consists of 4 steps: rule proposal, theorem proving, rule abstraction, and rule pruning. Let $n$ be the number of training examples, rule proposal and theorem proving are $O
> > (n)$. Rule abstraction is $\Omega(n^2)$ since we anti-unify each pair of rules. Rule pruning is NP-hard due to the use of MAX-SAT. Therefore, our system, in its current form, would indeed struggle with very large $n$. In order to scale to millions of examples, future work would have to reduce the complexity to $O(n)$.
> >
> > For rule abstraction, note that our pairwise anti-unification is only a design choice at the implementation level, rather than a conceptual necessity. In principle, all we need is a procedure for generating abstract rules from concrete ones. It is possible to develop more efficient algorithms for rule abstraction with linear or even sublinear runtime. They can also be based on machine learning, e.g., Cingillioglu and Russo [G].
> >
> > For rule pruning, one potential way to scale it up is through approximate MAX-SAT solvers. Similar to deep learning, MetaInduce does not necessarily require achieving the global minimum to work well. Another important direction to explore is learning in mini-batches. It avoids encoding the entire dataset as a large MAX-SAT problem, and instead solves multiple small MAX-SAT problems.
> >
> >
> >
> >
> > ### Experimental run time of MetaInduce
> >
> > Our SCAN experiments take 30 minutes to train on a laptop, which compares favorably with methods that use deep neural networks (e.g., 1 day on GPUs in Liu et al. [H]). On RuleTaker, our experiments take 2 days. It is impossible to directly compare with ProofWriter since the authors haven't released the code.
> >
> > Our  run time is noteworthy considering that deep neural networks are trained on software/hardware stacks (GPUs, cuDNN, PyTorch, etc.) highly optimized for them. And our method is at an early stage without nearly as much engineering effort for improving the run time efficiency.
> >
> >
> > * [H] Liu et al. "Compositional generalization by learning analytical expressions." NeurIPS, 2020.

---

### Decision · Program_Chairs · 2022-01-20

**Decision:**

Reject

**Comment:**

The paper describes a system for learning rules in a quasi-NL format: roughly Horn clauses where a predicate p(X1,...,Xk) is replaced by a natural language pattern interleaving ground tokens and variables.  The method is to propose ground sentences - using one of several task-specific approaches - use anti-unification of pairs to variabilize, and then find a minimal theory from these proposed pairs by reduction to maxsat.

Pros:
 - QNL is a neat idea, and makes symbolic rule-learning possible to some NLP tasks
 - The use of maxsat is novel in rule-learning AFAIK

Cons:
 - Unification is a highly simplified model of the NL task of cross-document co-reference
 - It's unclear if maxsat process will work in the presence of noise, or how well it scales
 - The datasets use clean text generated from templates or synthetic grammars
 - Experimentally, the generality of the system is not well demonstrated, because there are differences in how it is applied: eg a subset of short examples for scan, input engineering ($TRUE, $FALSE) for RuleTaker, plus the "heuristics for filtering invalid rules generated by anti-unification”
 - It's not clear if this work really speaks to the main "point" of the SCAN and RuleTaker datasets.  These are both the kind tasks that symbolic systems would be expected to do well, and are used as ANN benchmarks because ANNs perform in unexpected ways: worse than one would expect for SCAN, and better for RuleTaker.  They are important for understanding ANNs but I'm not certain what the research benefit is of using them for symbolic methods as a benchmark.